# Collagen Membrane Derived from Fish Scales for Application in Bone Tissue Engineering

**DOI:** 10.3390/polym14132532

**Published:** 2022-06-21

**Authors:** Liang Chen, Guoping Cheng, Shu Meng, Yi Ding

**Affiliations:** 1National Clinical Research Center for Oral Diseases, West China Hospital of Stomatology, Sichuan University, Chengdu 610041, China; clworkzone@126.com (L.C.); cgp19940611@foxmail.com (G.C.); dreamingsue@163.com (S.M.); 2State Key Laboratory of Oral Diseases, Sichuan University, Chengdu 610041, China; 3Department of Periodontology, West China College of Stomatology, Sichuan University, Chengdu 610041, China; 4Department of Periodontology, Peking University School and Hospital of Stomatology, Beijing 100081, China

**Keywords:** fish scale, collagen, hydroxyapatite, bone marrow mesenchymal stem cells, osteogenic differentiation

## Abstract

Guided tissue/bone regeneration (GTR/GBR) is currently the main treatment for alveolar bone regeneration. The commonly used barrier membranes in GTR/GBR are collagen membranes from mammals such as porcine or cattle. Fish collagen is being explored as a potential substitute for mammalian collagen due to its low cost, no zoonotic risk, and lack of religious constraints. Fish scale is a multi-layer natural collagen composite with high mechanical strength, but its biomedical application is limited due to the low denaturation temperature of fish collagen. In this study, a fish scale collagen membrane with a high denaturation temperature of 79.5 °C was prepared using an improved method based on preserving the basic shape of fish scales. The fish scale collagen membrane was mainly composed of type I collagen and hydroxyapatite, in which the weight ratios of water, organic matter, and inorganic matter were 20.7%, 56.9%, and 22.4%, respectively. Compared to the Bio-Gide^®^ membrane (BG) commonly used in the GTR/GBR, fish scale collagen membrane showed good cytocompatibility and could promote late osteogenic differentiation of cells. In conclusion, the collagen membrane prepared from fish scales had good thermal stability, cytocompatibility, and osteogenic activity, which showed potential for bone tissue engineering applications.

## 1. Introduction

Periodontitis is a chronic inflammatory disease that occurs in periodontal tissue and is the main cause of tooth loss in adults. Resorption of alveolar bone is one of the main pathological changes [1]. Traditional periodontal treatment methods, such as scaling and root planning, can only prevent the development of periodontal disease. However, alveolar bone regeneration is difficult to achieve [2]. Guided tissue/bone regeneration (GTR/GBR) is not only the main treatment of alveolar bone regeneration clinically but also the main means of bone augmentation during implant surgery. GTR/GBR is a technique that uses membranes as a barrier to keep gingival connective tissue from contacting the root surface, creating space, and guiding periodontal tissue or bone regeneration [3,4,5]. Collagen membranes, whose main component is type I collagen and whose most common source is mammals like porcine or cattle, are commonly used as barrier materials in the GTR/GBR.

Collagen is the most important structural protein in most soft and hard tissues of animals and humans, accounting for about 30% of the total proteins in mammals and playing a critical role in maintaining the biological and structural integrity of the extracellular matrix [6]. Collagen regulates cell morphology, adhesion, migration, and differentiation and has low immunogenicity, good biocompatibility, and biodegradability [7,8]. There are currently 29 different types of collagen identified, each with their own amino acid sequences, structures, and functions [9]. Type I collagen has the highest content of all of them, which is found in connective tissue such as skin, tendons, bones, ligaments, and cornea [10], accounting for more than 90% of the collagen in the human body [11]. Type I collagen is being studied for a variety of applications, including health products, cosmetics, hemostatic agents, wound healing, regenerative medicine, and drug delivery [10,12,13,14]. Skin and tendon tissues derived from mammals such as porcine or cattle are the main sources of type I collagen for biomedical applications [15]. However, mammalian-derived collagen has the following disadvantages: (1) safety: zoonotic diseases such as bovine spongiform encephalopathy and foot-and-mouth disease may be transmitted through mammals [16]; (2) religious restrictions [17]: Islam and Judaism do not use any porcine-derived products, while Hinduism does not use any cattle-derived products; (3) cost: mammalian collagen purification is difficult and expensive [11]. Fish-derived collagen is a potential substitute for mammalian collagen because of the low risk of zoonosis from fish to humans, the lack of religious restrictions, the low cost of fish scales and other aquatic scraps, and the fact that it is safe and easily obtainable [11]. As many as 50–70 percent of by-products were produced in the production and processing of fish products [18], with a large number of fish bones, swim bladders, and scales being discarded as leftovers. A growing number of researchers have focused on fish-derived collagen in recent years [19,20,21].

The bony fish scales are mainly composed of type I collagen fibers and hydroxyapatite, in which the protein content is 41–84% [22]. Hydroxyapatite enhances type I collagen fibers in fish scales, and the parallel collagen fibers overlap to form a multi-layer structure with a highly ordered three-dimensional structure [23,24]. Fish scales have a similar main composition to human bone and dentin [25], and their highly ordered natural multi-layer structure has good mechanical properties. Each year, about 49,000 tons of fish scale waste are produced during the processing of fish, accounting for 2% of the total weight of the fish [26]. It is necessary to use these fish industry waste to synthesize value-added materials for several different applications, such as energy storage and packaging [27]. If these waste by-products could be used, people could not only reduce pollution but also improve the utilization of fish products and economic benefits [28]. With regard to the research of grass carp (*Ctenopharyngodon idella*) scales, some scholars have tried to apply decellularized grass carp scales to artificial cornea [29,30]. In addition, a study has fabricated bone pins with decellularized grass carp scales and implanted them into the femur fractures of animals. Finally, it demonstrated that decellularized grass carp scales could be a promising implant material for bone repair [23]. Therefore, the purpose of this study is to prepare a decellularized fish scale collagen membrane (FS) from grass carp scales and analyze its structure and composition. Moreover, compared to Bio-Gide^®^ membrane (BG), a common commercial collagen membrane used clinically, we want to explore whether FS has good cytocompatibility and osteogenic activity and to further determine whether decellularized grass carp scales have the potential to be applied in bone tissue engineering.

## 2. Materials and Methods

All experiments were carried out in accordance with the ethical protocol and guidelines. Ethical approval was obtained from the ethical committees of the West China School of Stomatology, Sichuan University, and the State Key Laboratory of Oral Diseases (ethics code WCHSIRB-D-2020-220).

### 2.1. Preparation of Decellularized Fish Scale Collagen Membrane (FS)

The preparation method was roughly based on our previous research [30,31] and slightly improved as follows. In brief, about 100 grass carp scales with a diameter of more than 20 mm were selected and cleaned with distilled water to remove ash and impurities from the surface. Under the condition of continuously stirring the solution in the blender, the fish scales were treated with 80 mL of chloroform: ethanol = 1:1 mixed solution for 1 h to remove mucopolysaccharides, proteins, and fats and washed with 80 mL of 5% NaCl solution for 1 h to remove unnecessary proteins on the surface. The scales were then soaked in 100 mL of 10% EDTA solution (Solarbio, Beijing, China) for 4 h and then treated with a 0.5 mol/L acetic acid solution (Kelong, Chengdu, China) for 1 h, which was called decalcified fish scale. Finally, the decalcified fish scales were sprayed with a pepsin solution (Kelong, Chengdu, China) for 1 h to etch the surface. The treated fish scales were soaked in 75% ethanol for sterilization for about 12 h and then stored in aseptic PBS at 4 °C, which was called fish scale collagen membrane (FS). The whole preparation process was carried out at a temperature of 25 °C.

### 2.2. Structure, Composition, and Denaturation Temperature Analysis of FS

#### 2.2.1. Cell Nucleus Staining of FS

Cell nucleus staining was used to determine whether the material was decellularized or not. Decellularized FS seeded with cells were fixed for 20 min in 4% paraformaldehyde solution (Solarbio, Beijing, China) and then permeabilized for 5 min with 1% TritonX-100 (Solarbio, Beijing, China). After three PBS washes, the decellularized FS were incubated with 4′,6-diamidino-2-phenylindole dihydrochloride (Solarbio, Beijing, China) at room temperature for 20 min to label nuclei. The same procedure was used to treat the decellularized FS sample without being seeded cells. The DAPI stained cells were then observed with a fluorescence microscope (DFC7000T, LEICA, Wetzlar, Germany) at a wavelength of 350 nm.

#### 2.2.2. Morphology and Structure Analysis

After being freeze-dried at −20 °C for 24 h, FS was cut along the surface lines of fish scale with a disposable scalpel and fixed with 2.5% glutaraldehyde (Solarbio, Beijing, China) at room temperature for 20 min and at 4 °C for 2 h. After that, 50%, 70%, 90%, and 100% ethanol solutions were used sequentially for dehydration, with an interval of 10 min each time. After dehydration, the above samples were put into the CO_2_ critical point dryer for 1 h and sputter-coated with gold. The longitudinal section and surface morphology were observed and photographed with a scanning electron microscope (Inspect F, FEI, Hillsboro, OR, USA).

#### 2.2.3. Fourier Transform Infrared Spectroscopy (FTIR)

Fourier transform infrared spectroscopy (FTIR) of FS was performed with a Fourier transform infrared spectrometer (INVENIO R, Bruker, Billerica, MA, USA). The spectrometer was equipped with an attenuated total reflection (ATR) accessory with a germanium crystal. The FS were directly placed on the ATR crystal and measured under the following conditions: scan range of 450–4000 cm^−1^, 16 scans with a spectral resolution of 4 cm^−1^, and indoor temperature of 25 °C. Hydroxyapatite (Aladdin, Shanghai, China) was served as control.

#### 2.2.4. Thermogravimetric Analysis

Thermogravimetric analysis (TGA) of FS was performed with a thermogravimetric analyzer (TGA/DSC2, Mettler Toledo, Switzerland) under a nitrogen atmosphere. Heating temperature ranged from 30 °C to 1200 °C at a heating rate of 10 °C/min. The data were analyzed and graphed using Origin software.

#### 2.2.5. Differential Scanning Calorimetry (DSC)

The thermal stability of FS was evaluated with DSC. The 10–15 mg sample was sealed in an aluminum crucible and heated from 30 °C to 90 °C at a rate of 5 °C/min under a nitrogen atmosphere. The data were analyzed and graphed using Origin software.

### 2.3. Culture and Identification of Bone Marrow Mesenchymal Stem Cells (BMSCs)

#### 2.3.1. Culture of BMSCs

Two male SD rats, 3–4 weeks old, SPF grade, weighing 200 g, were purchased from Chengdu Dashuo Experimental Animal Center, and rat BMSCs were isolated and cultured as described previously [30]. In brief, the bilateral tibia and femur were aseptically separated under sterile conditions and bilateral epiphyses were cut off. The medullary cavity was rinsed and bone marrow collected using a 5 mL syringe with α-MEM (Gibco, New York, NY, USA) medium containing 10% fetal bovine serum (Gibco, New York, NY, USA) and 1% penicillin-streptomycin solution (Hyclone, Logan, UT, USA). The cell collection was centrifuged at 1000 r/min for 5 min and was cultured at 37 °C in a humidified atmosphere of 95% air and 5% CO_2_. After 24 h, the medium was changed for the first time, and then, it was changed every 2–3 days. The cell proliferation and growth status were observed with the inverted microscope (Olympus, Tokyo, Japan). The third passages of BMSCs were selected for subsequent experiments.

#### 2.3.2. Flow Cytometry Analysis

BMSCs were identified with flow cytometry. Cells were stained with the following antibodies: CD29-APC (BioLegend, San Diego, CA, USA), CD45 PerCP/Cyanine 5.5 (BioLegend, San Diego, CA, USA), CD90-FITC (eBioscience, San Diego, CA, USA). After being incubated for 1 h, cells were washed with PBS 3 times before being analyzed on a BD FACSCanto II cytometer. The experimental data were analyzed with FlowJo 10.0.7 software (Franklin Lakes, NJ, USA).

#### 2.3.3. Osteogenic and Adipogenic Differentiation

To induce osteogenic differentiation, an osteogenic differentiation medium was prepared, which contained α-MEM, 10% fetal bovine serum, 50 μmol/L ascorbic acid (Sigma, Ronkonkoma, NY, USA), 0.1 μmol/L dexamethasone (Solarbio, Beijing, China), 10 mmol/L β-glycerophosphate (Sigma, Ronkonkoma, NY, USA), and 1% penicillin-streptomycin solution. Cells cultured with regular culture medium were used as a control. After 7 days of differentiation, BMSCs were fixed in a 4% paraformaldehyde (Solarbio, Beijing, China) for 30 min and stained with BCIP/NBT ALP color development kit (Beyotime, Shanghai, China). After 21-day differentiation, BMSCs were stained with Alizarin red S staining solution (Solarbio, Beijing, China). After washing with ddH_2_O, the cells were observed under an inverted microscope.

Adipogenic differentiation medium consisted of DMEM (Gibco, New York, NY, USA), 10% fetal bovine serum, 0.5 μmol/L 3-isobutyl-1-methylxanthine (Solarbio, Beijing, China), 0.5 μmol/L dexamethasone, 10 μg/mL insulin (Solarbio, Beijing, China), 50 μmol/L indomethacin (Solarbio, Beijing, China), and 1% penicillin-streptomycin solution. BMSCs were cultured in an adipogenic differentiation medium or regular culture medium, and the medium was changed every 2–3 days. After 14 days, the cells were fixed in a 4% paraformaldehyde for 30 min and incubated for 10 min in an Oil Red O (Solarbio, Beijing, China) working solution. After thorough washes with a 75% ethanol solution, the cells were observed under an inverted microscope for the presence of lipid droplets, which were stained red.

### 2.4. Cytocompatibility of FS and Bio-Gide^®^ Membrane (BG)

#### 2.4.1. Cell Adhesion

FS and BG were trimmed with a punch (12 mm in diameter) and placed onto 24-well culture plates with the striated side of FS facing up and the loose side of BG facing up. BMSCs counting 1 × 10^5^ were seeded on both materials and cultured for 1 h and 6 h at 37 °C in a CO_2_ incubator. Cells on the materials were fixed with 2.5% glutaraldehyde (Solarbio, Beijing, China) at room temperature for 20 min and at 4 °C for 2 h. Then ethanol of different mass fractions was dropped onto the materials to dehydrate the cell samples. Finally, SEM was used to observe the morphology of BMSCs.

#### 2.4.2. Cell Proliferation

The effect of cell proliferation was analyzed with the CCK8 assay. The eluate from FS and BG was prepared according to ISO10993-5-2009 [32]. Briefly, the aseptic 25 mm × 25 mm BG was incubated with 31.25 mL α-MEM medium without fetal bovine serum (FBS) for 24 h at 37 °C in a CO_2_ incubator, where the extraction rate was 20 mm^2^/mL. The eluate extraction was collected and stored for further analysis at 4 °C. The eluate of FS was prepared with the same extraction rate. BMSCs were seeded into a 96-well plate at 1 × 10^3^ cells per well with a 100 μLα-MEM medium containing 10% FBS and cultured in an incubator at 37 °C with 5% CO_2_. After a 24 h culture, the medium was replaced with 100 μL FS or BG eluate containing 10% FBS. As blank controls, cells cultured in -MEM medium containing 10% FBS were used. CCK8 was added (10 μL per well), and the plates were incubated at 37 °C for 2 h. The absorbance values at OD 450 nm were measured on the 1st, 3rd, and 5th day using a Multiskan™ FC microplate reader (Thermo Fisher, Waltham, MA, USA).

### 2.5. Osteogenic Activity of FS and BG

#### 2.5.1. ALP Staining and ALP Activity Assay

FS and BG were trimmed with a punch (12 mm in diameter) and placed onto 24-well culture plates with the smooth side of FS facing up and the dense side of BG facing up. The same size aseptic glass slides were used as the control group. A custom polytetrafluoroethylene ring was used to press the materials around the edges of the plates, causing them to sink to the bottom. Four parallel samples were set up in each group. BMSCs counting 5 × 10^4^ were seeded onto the plates with pre-placed materials and cultured for 24 h at 37 °C in a CO_2_ incubator. Then, the medium was replaced with an osteogenic differentiation medium. ALP staining and ALP activity were determined after 7 days using the BCIP/NBT ALP color development kit and the ALP assay kit (Beyotime, Shanghai, China).

#### 2.5.2. Relative Quantitative Assay of Mineralized Deposits

As previously stated, the same number of materials were placed onto 24-well culture plates in the same way. After 21-day osteogenic differentiation, BMSCs were stained with Alizarin red S staining solution, and stained materials were washed with distilled water several times to remove stain residue. The stained samples were then eluted with 10% cetylpyridinium chloride (Aladdin, Shanghai, China). The absorbance values at OD 562 nm were measured using a Multiskan™ FC microplate reader.

#### 2.5.3. ELISA Analysis

ELISA was performed to analyze osteocalcin expression of BMSCs on the different materials. After 21-day osteogenic differentiation, supernatants from cell cultures were collected and detected using the Rat OC/BGP (osteocalcin) ELISA Kit (Elabscience, Wuhan, China). Finally, a Multiskan™ FC microplate reader was used to measure the absorbance values at OD 450 nm.

#### 2.5.4. RT-qPCR Analysis

Osteogenesis-related gene expression quantification was performed with RT-qPCR. FS, BG, and glass slide control were placed onto 24-well culture plates in the same way as previously (*n* = 3). BMSCs counting 5 × 10^4^ were seeded onto the plates and osteogenic differentiation induced for 7 days and 21 days at 37 °C in a CO_2_ incubator. Cells were collected after digestion with trypsin (Gibco, New York, NY, USA), and total RNA was extracted using an RNA extraction kit (Bioteke, Beijing, China) at the indicated times. Total RNA samples were reverse transcribed to cDNA using a PrimeScriptTMRT reagent Kit with gDNA Eraser (Takara, Kusatsu, Japan). Real-time quantitative PCR was performed with TB Green^®^ EX TaqTM II (Takara, Kusatsu, Japan) in a QuantStudio 6 instrument (Thermo Scientific, Waltham, MA, USA). The primer sequences are shown in Table 1.

### 2.6. Statistical Analysis

SPSS 19.0, GraphPad Prism 8 (San Diego, CA, USA), and Origin 8.5 software (Northampton, MA, USA) were used for data analysis. Statistical analysis was performed by one-way ANOVA. Statistical significance was determined using Bonferroni correction when the variances were homogeneous, and Tamhane’s T2 test was used for non-homogeneous variances. A *p* < 0.05 was considered statistically significant (NS, *p* > 0.05, * *p* < 0.05, ** *p* < 0.01, *** *p* < 0.001).

## 3. Results

### 3.1. Morphology and Structure of FS

DAPI staining showed that FS was successfully decellularized, and no nuclei were found on the FS (Figure 1B). In contrast, after being seeded the cells, the nucleus appeared on the FS (Figure 1A).

FS is oval and transparent, about 25 mm in diameter with the basic shape of fish scale (Figure 2C). SEM analysis indicates that there were “petal-like” radial ridges and no pores on the surface of FS. The upper surface of FS (Figure 2D) is irregular with curved ridges. On the lower surface (Figure 2F), parallel circular ridges and longitudinal grooves are seen. The lateral surface of FS (Figure 2E) is more uniform with parallel circular ridges forming concentric arcs with the scale focus in the center. FS is a multi-layer structure with type I collagen fibers arranged in parallel and some spherical or cubic crystals embedded between each layer of collagen fibers, as shown in the longitudinal section (Figure 1A and Figure 2A,B).

### 3.2. Composition and Denaturation Temperature of FS

FTIR analysis (Figure 3A) showed five characteristic bands of type I collagen [33]: amide A (3292 cm^−1^), amide B (2933 cm^−1^), amide I (1630 cm^−1^), amide II (1550 cm^−1^), and amide III (1236 cm^−1^). In addition, the weak absorption peak at 3076 cm^−1^ is related to the C-N stretching vibration of amide B, while the 1382 and 1336 cm^−1^ absorption peaks are related to the rocking vibration of -CH_2_. It was reported that collagen has a complete triple helix secondary structure when the absorption ratio A1235/A1450 cm^−1^ of collagen is about 1.0 [34]. The FS prepared in this study has a ratio of A1235/A1450 cm^−1^ of 1.016, indicating that the collagen in the material has a complete triple helix structure. There are several absorption peaks of FS at 1029–1078 cm^−1^ and 524–623 cm^−1^, corresponding to the PO_4_ of apatite crystal. As shown in the figure, the same absorption peaks are present in the absorbance spectra of hydroxyapatite reference. The result suggests that FS is mainly composed of type I collagen and hydroxyapatite.

The thermogravimetric analysis (Figure 3B) revealed that there were three areas of mass loss: 30–178 °C, 178–650 °C, and 650–1200 °C. The first stage of mass loss of 30–178 °C was related to the evaporation of water molecules. The second stage of 178–650 °C was due to the decomposition of organic matter. When the temperature was raised to 1200 °C, the mass no longer changed, and the thermal decomposition of inorganic matter was completed in the third stage of 650–1200 °C. The mass loss curves demonstrated that the mass ratios of water, organic matter, and inorganic matter in FS are 20.7, 56.9, and 22.4%, relatively. Because type I collagen is the organic phase in FS and hydroxyapatite is the mainly inorganic phase, the ratio can be approximately estimated as the mass ratio of water, type I collagen, and hydroxyapatite.

DSC results (Figure 3C) showed a typical endothermic peak at 79.5 °C associated with irreversible denaturalization of collagen protein, indicating that the denaturation temperature of FS was 79.5 °C.

### 3.3. Culture and Identification of BMSCs

BMSCs were spindle-shaped and grew in a swirl shape. As shown from the P0-P3 generation cells (Figure 4A), with the increase of passage, the miscellaneous cells gradually decreased, and BMSCs were naturally purified by passage. Flow cytometry analysis suggested that BMSCs of the primary culture were mainly positive for CD90 and CD29 and negative for CD45 (Figure 4B). Cells after 7 days of osteogenic differentiation showed increased ALP staining compared with negative control cells, while cells cultured in the osteogenic medium for 21 days exhibited obvious amorphous calcium deposits by alizarin red staining compared with negative control cells (Figure 4C). The results of Oil Red O staining demonstrated numerous lipid droplets within BMSCs after adipogenic differentiation (Figure 4C). BMSCs expressing specific surface antigens were plastic-adherent and possessed multipotent differentiation potential, which meets the minimal criteria for defining MSCs [35].

### 3.4. Cytocompatibility of FS and Bio-Gide^®^ Membrane (BG)

Cell adhesion on the materials was observed by SEM (Figure 5). BMSCs on the FS exhibited abundant pseudopods and filopodia at 1 h, and the cells spread fully and extended further pseudopodia at 6 h (Figure 5A). As a contrast, BMSCs were ellipsoidal with no obvious pseudopodia and adhered to the collagen fibers of porous BG, which were the same at 1 h and 6 h (Figure 5B).

The results of the CCK8 assay (Figure 6) showed that the cell proliferation in the BG group and FS group was better than that in the control group. The number of cells in the FS group was even higher than that in the BG group on the 5th day, indicating that both FS and BG can promote cell proliferation, and the effect of FS on cell proliferation is stronger.

### 3.5. Osteogenic Activity of FS and BG

ALP staining of osteogenic differentiation at 7 d (Figure 7A) showed that there were obvious dark blue granular in the glass slide control, BG, and FS groups. Because BG was an opaque material, while glass slides and FS were relatively transparent, ALP staining in the BG group was darker under a stereoscopic microscope. ALP activity assay showed that there was no significant difference among the three groups (Figure 7B).

After 21-day osteogenic differentiation (Figure 8A), mineralized deposits in the FS group were significantly more than those in the control group (*p* < 0.001) and the BG group (*p* < 0.01), while there was no significant difference between the BG group and the glass slide control group (*p* > 0.05). Similarly, in the results of ELISA (Figure 8B), osteocalcin in the supernatant of cell culture in the FS group was significantly higher than that in the control group (*p* < 0.001) and the BG group (*p* < 0.05), and there was no difference between the latter two groups (*p* > 0.05).

At 7 days, there was no significant difference in the expression of osteogenesis-related genes RUNX2, OCN, and OPN among the three groups (Figure 9), which was consistent with the results of the ALP activity assay. After 21-day osteogenic differentiation, the gene expression of RUNX2 (*p* < 0.05), OCN (*p* < 0.05), and OPN (*p* < 0.01) in the FS group was significantly higher than that in the control group, and the expression of OCN (*p* < 0.05) and OPN (*p* < 0.01) in the FS group was significantly higher than that in the BG group. The above results of osteogenic activity suggested that FS did not significantly promote the osteogenic differentiation of BMSCs in the early stage (7 days), but it could promote the late osteogenic differentiation (21 days).

## 4. Discussion

Fish-derived collagen is considered to be a safer substitute for mammalian collagen, but the denaturation temperature of fish-derived collagen is usually lower than that of mammals [11,19], which limits its biomedical application to some extent. The denaturation temperature of collagen in different fish varies due to differences in habitat and body temperature. Collagen extracted from freshwater fish has a higher denaturation temperature than collagen extracted from marine fish [36]. Because freshwater fish collagen contains more imino acids (hydroxyproline and proline) than marine fish collagen, the denaturation temperature of collagen rises with the increase of imino acid content [37]. At present, the most studied freshwater fish is tilapia, which is a tropical freshwater bony fish with a large scale of cultivation, and the denaturation temperature of tilapia collagen is about 37 °C [38], which is equivalent to the human temperature. The grass carp (*Ctenopharyngodon idella*) was selected in this study, which belongs to the freshwater bony fish as tilapia. According to the *China Fisheries Statistical Yearbook 2020*, the annual output of grass carp was the highest among fish, about 5.53 million tons, while that of tilapia was about 1.64 million tons. The denaturation temperature of grass carp collagen was reported to be around 35–39 °C [39,40], similarly high to that of tilapia. In addition, adult grass carp is larger than tilapia generally, and the fish scales of large grass carp can reach more than 25 mm in diameter, while only about 10 mm for tilapia. In terms of utility, larger fish scales can be used to make larger collagen membranes. In this study, the denaturation temperature of FS is 79.5 °C, which is much higher than the human body temperature. It is possible that the high denaturation temperature of FS is due to the fact that the spatial structure of fish scales was preserved during the preparation, and collagen fibrils instead of collagen molecules were obtained. Collagen fibrils are more stable than extracted collagen molecules because intermolecular and intramolecular interactions stabilize the triple helix structure of collagen, and the denaturation temperature of type I collagen in collagen fibrils is usually higher than that in solution [41]. Moreover, hydroxyapatite in FS can improve the structural stability of collagen [42].

To extract collagen from fish scales, it is commonly solubilized in organic acid (usually acetic acid), but the yield of acid solubilized collagen is low, and the use of pepsin can increase the extraction yield of collagen and reduce its antigenicity [43]. In most studies on fish scale collagen, the combined extraction method of acid and pepsin was used [19]. Then, collagen fibers scaffolds for tissue engineering were prepared by a freeze-drying methodology or to be physically or chemically modified [44,45]. However, the natural multilayer and compact structure of fish scales would be destroyed during this collagen extraction process. In addition, decellularization is the key process in the preparation of biomedical materials derived from fish scales, as incomplete decellularization can result in an allergic reaction when the fish scale is implanted [23]. The study’s novelty is that the cells and impurities such as ash, keratin, and fat on the surface of fish scales were removed, and FS was successfully prepared by decellularization and decalcification on the basis of preserving the basic shape and spatial structure of fish scales. The natural multilayer structure of fish scales was not damaged, and the collagen fibrils and native mechanical strength have remained. This preparation of decellularization and decalcification is similar to the method of preparing artificial cornea using fish scales [46,47,48], but the preparation process in this study is simpler and more efficient.

General observation and the result of SEM showed that FS was a multilayer material with the basic shape and structure of fish scale, which is very similar to the fish scales of *Carassius auratus* [25], *Cyprinus carpio* [49], and tilapia [48]. Through the analysis of FTIR spectra, the results showed that the FS prepared from grass carp was mainly composed of type I collagen and hydroxyapatite, which was consistent with the fish scales such as *Pagrus major* [50], *Lates calcarifer* [22], *Carassius auratus* [25], *Cyprinus carpio* [49], *Megalops Atlanticus* [51], and tilapia [52]. After the preparation and treatment of fish scales, the weight ratio of water, organic matter, and inorganic matter is similar to that of decalcified grass carp scale reported before [23], and the mechanical properties of this decalcified fish scale material are significantly higher than that of pure collagen scaffold material [29]. For the mechanical properties, compared to Bio-Gide^®^ membrane, commonly used in the GTR/GBR, which is a kind of non-cross-linked, porcine-derived type I and III collagen membrane with a double-layer structure [53], our previous research [31] found that the thickness of FS is about 0.34 ± 0.05 mm, slightly lower than that of BG 0.44 ± 0.03 mm, but its tensile strength is obviously higher than BG in both dry and wet conditions, and FS degraded more slowly compared with BG.

Collagen derived from fish scales has been shown to have good cytocompatibility in a number of studies [23,54,55,56,57,58,59,60]. Similarly in this research, both FS and BG could preserve cellular adherence and well support the growth of BMSCs, indicating their good cytocompatibility. ALP is an important membrane-bound enzyme expressed in the early stage of cell osteogenesis [61,62]. The results of ALP activity assay and osteogenesis-related gene expression at 7 days showed that both FS and BG could not promote early osteogenic differentiation. Relatively, OCN is an essential late marker of osteogenic differentiation [63,64] for the formation of minerals, and the mineralization of the extracellular matrix is the characteristic of late osteogenic differentiation [65]. According to the results of alizarin red staining and ELISA, the mineralized deposits and OCN expression of BMSCs on the FS were higher than those on the BG and the glass slide control. Osteopontin (OPN) is a kind of extracellular matrix glycoprotein, which plays an important role in the process of bone remodeling [66] and is a marker of the intermediate stage of osteoblast differentiation [67]. RUNX2 is a key transcription factor required for osteoblast differentiation [68]. The expression of osteogenesis-related genes in the late stage (21 d) of osteogenic differentiation was significantly higher than that in the BG group and the glass slide control group, indicating that FS could promote late osteogenic differentiation.

In this study, it was found that FS was better than BG in promoting osteogenesis in vitro, and one of the reasons may be the existence of hydroxyapatite (HA). The main inorganic component of human bones and dentin is HA. It has been reported that collagen membrane mixed with HA had better mechanical properties [16], and HA could not only promote the protein adsorption and stimulate the osteogenic differentiation of stem cells [69] but also endow biomaterials with good bioactivity and osteoconductivity [70]. Parmaksiz et al. [71] used HA microparticles and polycaprolactone as binders to assemble bovine intestinal submucosa into multilayer scaffold material. The morphological observation results showed that it had a uniform multilayer pore structure and uniform distribution of HA microparticles. Mechanical tests showed that it had good mechanical stability, and cell experiments showed that it had a good osteoinductive effect on BMSCs even without any external osteogenic inducers. At present, the mechanism of HA promoting osteogenic differentiation is not clear, which may be the action of calcium ions or calcium phosphates from HA [62,72]. Studies have shown that calcium ions are involved in various physiological activities and electrical signal transduction of cell osteogenic differentiation [73], and biomaterials containing calcium phosphate can induce osteogenic differentiation of stem cells through adenosine signaling [74]. On the other hand, the osteogenic activity of FS may also be derived from the collagen of fish scales. Matsumoto et al. [59] found that fish scale collagen can promote the osteogenic differentiation of human MSCs; Studies by Hsu et al. [58] have shown that fish scale collagen can promote chondrogenic cartilage differentiation of human MSCs, and its effect is stronger than that of porcine collagen; Liu et al. reported that hydrolyzed fish scale collagen can directly induce osteogenic differentiation of rat BMSCs [57] and human periodontal ligament cells [75], without the use of any additional inducing reagent, and significantly up-regulate the expression of osteogenic markers (RUNX2, ALP, OPN, and OCN) [57].

It has been reported that the porosity and morphology of materials will affect the effect of bone regeneration in vivo [76]. Generally, the osteogenic effect of perforated barrier membrane is better than that of non-perforated barrier membrane [77]. In addition, macroporous membranes facilitated greater bone regeneration compared with microporous membranes [78]. Due to the differences in morphology and pores between different materials, the materials will be affected by the immune response after being implanted in vivo [3], and it is difficult to compare the effects of different materials on osteogenic differentiation. In the part of the osteogenic differentiation experiment, not only FS and BG were compared but also a glass slide group was set up as a negative control, in which both FS and glass slide had no pores. Considering that there were no obvious pores on the dense side of BG, we put the dense side of BG upward, which can minimize the influence of pores on osteogenic differentiation. The results demonstrated that after removing the effect of pores, BG could not promote osteogenic differentiation of BMSCs (7 d and 21 d), while FS could promote osteogenic differentiation at the late stage (21 d) in vitro. In our previous research [31], FS and BG were implanted into the rat cranial bone defect. The results showed that there was no significant difference in the amount of new bone formation between the FS and BG groups after 4 weeks (*p* > 0.05). At the 8th week after operation, BG was completely degraded, and FS remained partially, and the newly formed bone completely filled in the bone defects in the BG group, which was higher than that in the FS group (*p* < 0.05). The results of osteogenic differentiation in previous animal experiments are inconsistent with this research in vitro, which may be caused by the effect of porosity and speed of degradation. FS has no obvious pores and may not be beneficial to the blood–gas exchange of tissues and cells after implantation in vivo. Moreover, FS degraded more slowly than BG in vivo, which does not match the rate of rat bone regeneration, and the residual FS membrane would occupy the space of newly formed bone. Taken together, the results of our previous animal experiment and this research in vitro suggest that FS may have additional osteoinductive properties compared with BG besides the osteoconductivity. However, because FS has no pores and no appropriate degradation rate for rats, its osteogenic effect in vivo is not as good as that in vitro.

To sum up, BG had good cytocompatibility, but the dense side of BG had no significant effect on promoting osteogenic differentiation of BMSCs on the 7th and 21st day. The cytocompatibility of FS was excellent, and its effect on cell proliferation was superior to that of BG (*p* < 0.05). In addition, FS had no obvious effect on promoting osteogenic differentiation at the early stage (7 d), but it could promote cell osteogenic differentiation at the late stage (21 d), indicating its good cytocompatibility and osteogenic activity, as well as the potential to be used in bone regeneration. In order to achieve good bone regeneration, it is necessary to improve the porosity, pore size and degradation rate of FS in the future so that, when used in vivo, it can not only act as a barrier for epithelium and connective tissue but also benefit blood–gas exchange between tissue and cells on both sides of the membrane.

## Figures and Tables

**Figure 1 polymers-14-02532-f001:**
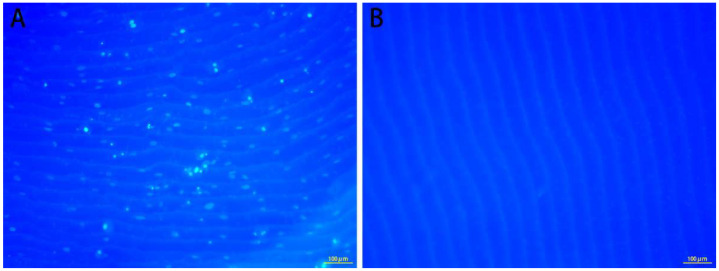
DAPI staining of FS with cells (**A**) and decellularized FS (**B**).

**Figure 2 polymers-14-02532-f002:**
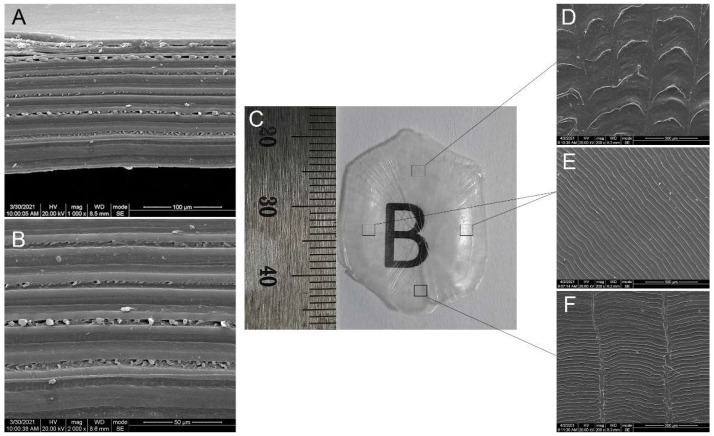
SEM images of surface (**D**–**F**) and longitudinal section (**A**,**B**) of FS (**C**).

**Figure 3 polymers-14-02532-f003:**
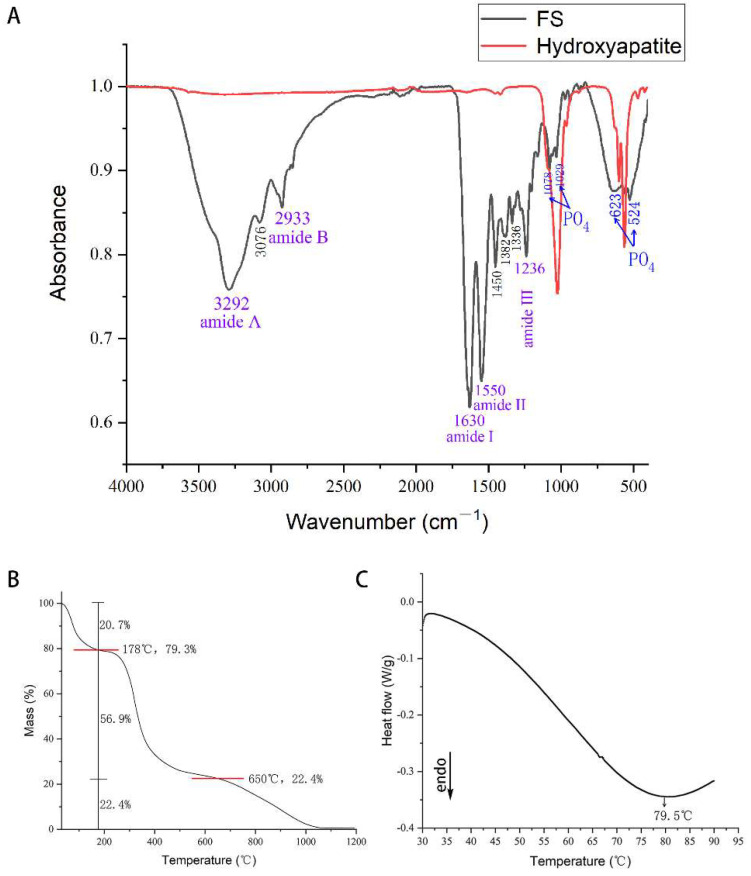
FTIR (**A**), thermogravimetric analysis (**B**), and DSC (**C**).

**Figure 4 polymers-14-02532-f004:**
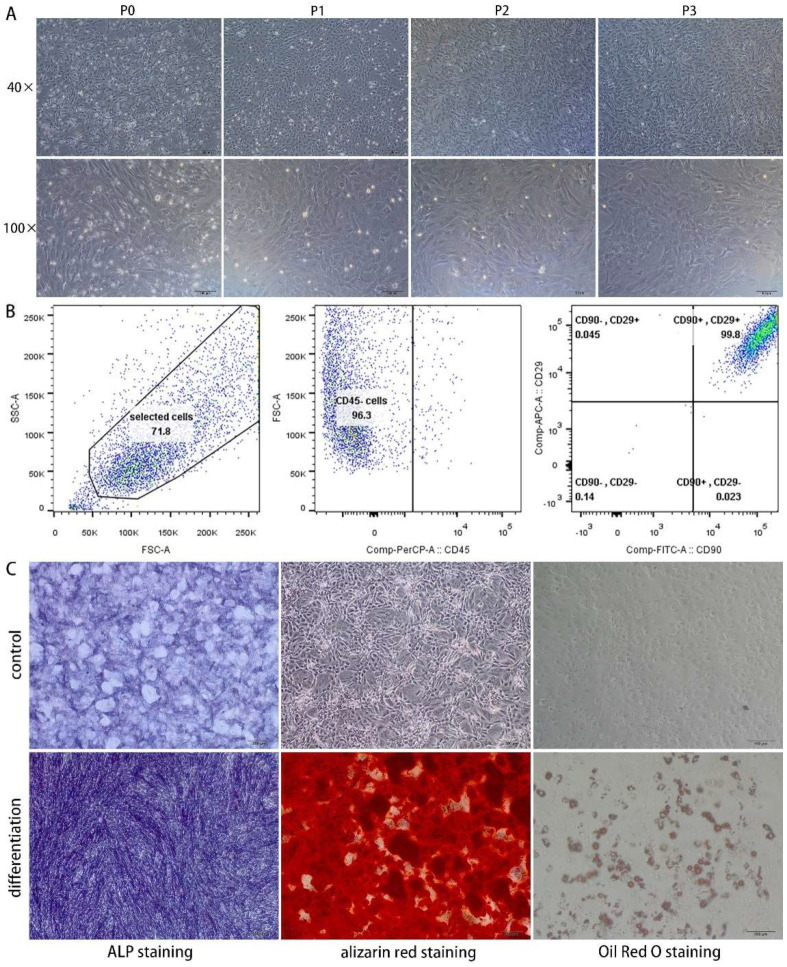
Morphology of BMSCs (**A**), flow cytometry analysis (**B**), and multipotent differentiation (**C**).

**Figure 5 polymers-14-02532-f005:**
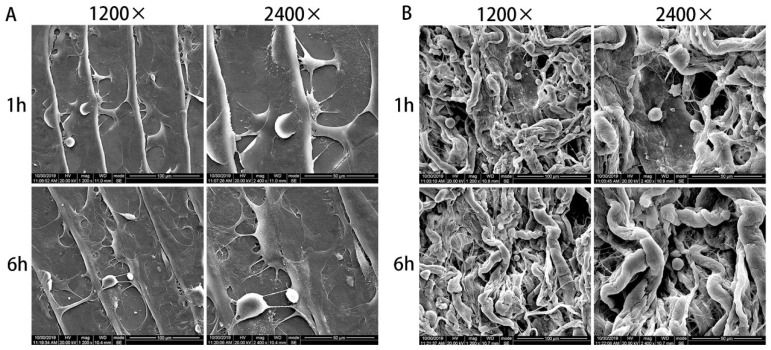
Cell adhesion of BMSCs on the FS (**A**) and BG (**B**).

**Figure 6 polymers-14-02532-f006:**
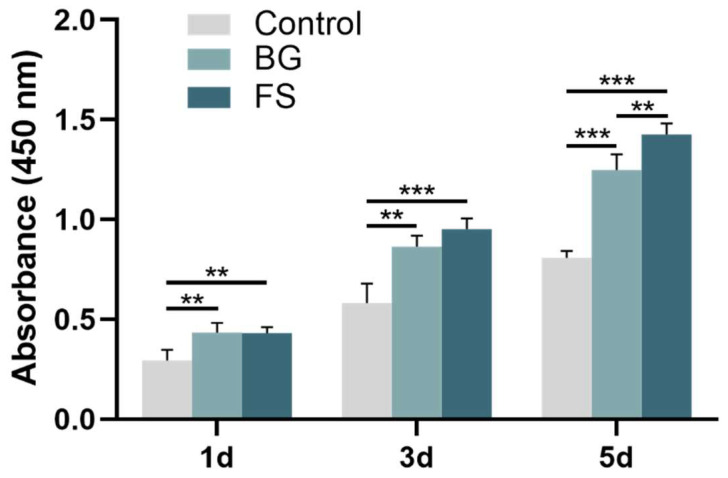
CCK8 assay of control, BG and FS group (** *p* < 0.01, *** *p* < 0.001).

**Figure 7 polymers-14-02532-f007:**
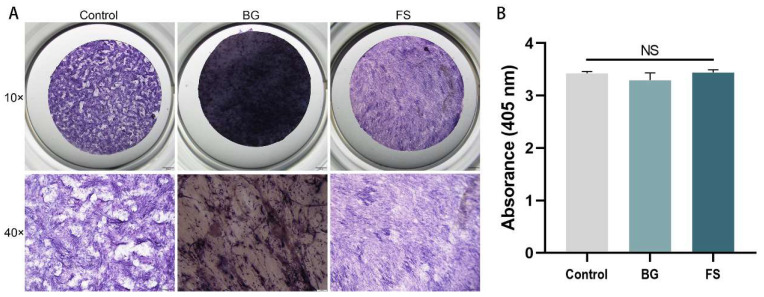
ALP staining (**A**) and ALP activity assay (**B**) of glass slide control, BG and FS group (NS, *p* > 0.05).

**Figure 8 polymers-14-02532-f008:**
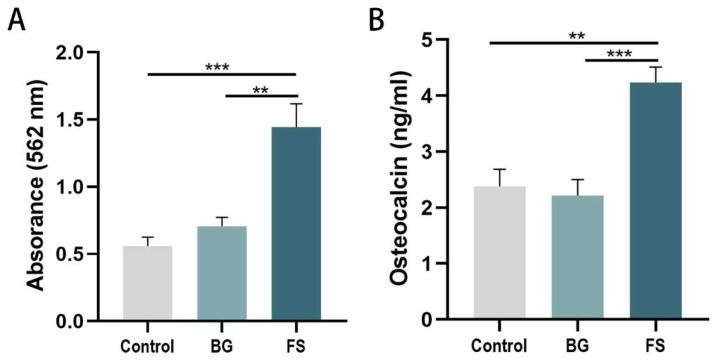
Relative quantitative assay of mineralized deposits (**A**) and ELISA analysis (**B**) of glass slide control, BG and FS group (** *p* < 0.01, *** *p* < 0.001).

**Figure 9 polymers-14-02532-f009:**
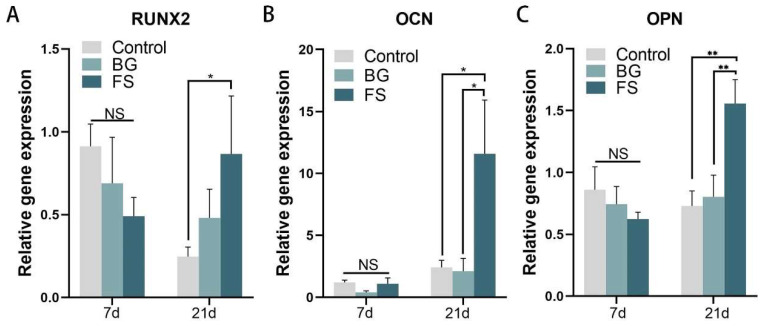
The relative expression of RUNX2 (**A**), OCN (**B**), and OPN (**C**) in the glass slide control, BG and FS group (NS, *p* > 0.05, * *p* < 0.05, ** *p* < 0.01).

**Table 1 polymers-14-02532-t001:** Primer sequences.

Gene	Primer Sequence (5′–3′)
OCN	Forward: CGCCAGGGTGAAGAACTA
	Reverse: TACGCTGTGGAAGCCAA
OPN	Forward: CGGAGACCATGCAGAGA
	Reverse: CGTAAGCCAAGCTATCACC
RUNX2	Forward: TCGGAAAGGGACGAGAG
	Reverse: TTCAAACGCATACCTGCAT
ACTB	Forward: CCTCACTGTCCACCTTCCA
	Reverse: GGGTGTAAAACGCAGCTCA

## Data Availability

This study did not report any data.

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
