# Peer review of "Collagen Membrane Derived from Fish Scales for Application in Bone Tissue Engineering"

_polymers, 2022, doi:10.3390/polym14132532_

Round 1

Reviewer 1 Report

Dear authors,

I appreciate the complex biological evaluation of the proposed acellular fish scale collagen membrane for its prospective application, but the manuscript needs to revise several major aspects.

Please rewrite the Abstract, since it appears disjointed in places. Try to present more clearly the rationale and motivation for your study, avoid abbreviations or write the name in detail.

The topic of the sentences seems twisted and hard to follow. For instance „Collagen membranes, the main component of which is type I collagen, are most commonly used in GTR/GBR, and the most common source is mammals such as porcine or cattle.”. Try to be more concise and use shorter phrases.

The novelty of the study is not reflected. In the Introduction, I suggest presenting similar previous literature (if any) regarding the usage of acellular fish scale collagen membrane from grass carp scales.

Please be consistent in the terminology used (acellular/ decellularized). Decellularization is defined as the process to create an acellular component. Maybe the title for section 2.1 should be „Decellularization of fish scale collagen membrane”.

It is hard to follow the Materials and Methods part. In section 2.1 please explain the role of each solvent/step performed for the decellularization of the fish scale.  Moreover, section 2.3.1 is referred to in section 2.2.1 („either with or without cells from 2.3.1”). For the morphology analysis is very ambiguous to state that „gold was sprayed 1 hour later”. Also, the samples are not „photographed” by scanning electron microscope, the obtained image is not a real one, but a representation based on the energies of collected electrons.

What did you mean by „The proportion of organic and inorganic components in FS was detected by a thermogravimetric analyzer”? The mass loss in this type of analysis is not only attributed to organic compounds but also to volatile water or decomposing inorganic salts (especially since the temperature range is 30- 1200 ℃).

Please check the entire manuscript for English, editing, and typing errors.

Reviewer 2 Report

The paper presents an experimental work on the preparation of a collagen membrane from fish scales with good thermal stability, cytocompatibility and osteogenic activity. The results are very interesting and promising overcoming the limitations of marine collagen. I suggest to improve the scientific quality of this good work with a deeper explanation of the strategies to keep the natural multilayer and compact structure of fish scales.

1.      In the Introduction, it should be better underline the importance of fish industry waste for the synthesis of value-added materials for several different applications including energy storage and packaging. Cite for example Energies 2021, 14, 7928. https://doi.org/10.3390/en14237928 and other very recent works on this subject.

2.      Page 2, row 65: Please add recent review papers published on Polymers journal focused on the recovery of collagen from fish waste.  

3.      Please add a scheme summarizing the serval steps for the preparation of decellularized fish scale collagen membrane. Moreover, in the experimental section, explain the reasons of the different treatments

4.      In the Experimental section the authors should add details about the motivations of the single steps or treatments. For example, why the surfactant TritonX-100 is used before the staining? Which is the composition of DAPI solution? Which is the content of DAPI solution used for Cell nucleus staining of FS? Why is used glutaraldehyde?

5.      How the sample was prepared to be investigated by the FTIR?

6.      DSC results: please add the endothermic direction on Figure 3C. The thermogram up to 90 °C is not nice to see. Please replace Fig 3C with one reporting a scan up to 200-250°C in order to see the end of the denaturation process. It is known that a limit of marine collagen is the low denaturation temperature. The authors obtained a high value of denaturation temperature, equal to 79.5 °C. Please compare it with the values of other marine and mammalian collagen reported in the literature.

7.      Please underline which innovation in the protocol extraction enabled to keep the natural multilayer and compact structure of fish scales, achieving high yield and denaturation temperature. Can the presence of hydroxyapatite considered a sort of nanostructured structure?

8.      Give an estimation of the yield of collagen extracted from fish scales and the overall duration of the process

Round 2

Reviewer 1 Report

Dear authors,

I appreciate your efforts in improving the manuscript. In the Abstract, please add the measuring unit for "denaturation temperature of 79.5".

Reviewer 2 Report

The authors have addressed nearly all my comments improving the quality o the paper.